# LEARNING TO (LEARN AT TEST TIME)

## ABSTRACT

We reformulate the problem of supervised learning as learning to learn with two nested loops (*i.e.* learning problems). The inner loop learns on each individual instance with self-supervision before final prediction. The outer loop learns the self-supervised task used by the inner loop, such that its final prediction improves. Our inner loop turns out to be equivalent to linear attention when the inner-loop learner is only a linear model, and to self-attention when it is a kernel estimator. For practical comparison with linear or self-attention layers, we replace each of them in a transformer with an inner loop, so our outer loop is equivalent to training the architecture. When each inner-loop learner is a neural network, our approach vastly outperforms transformers with linear attention on ImageNet from $224 \times 224$ raw pixels in both accuracy and FLOPs, while (regular) transformers cannot run.

## 1 INTRODUCTION

Test-time training (TTT) is an algorithmic framework for machine learning. The core idea is that each test instance defines its own learning problem, with its own target of generalization (Sun et al., 2020). Since the test instance comes without its label, TTT is performed with a self-supervised task such as reconstruction. Performance should improve on this particular instance for the self-supervised task, because that is the objective optimized by TTT. But will such a process lead to better performance for the main task we actually care about?

If improvement for a self-supervised task transfers to a given main task, we say the two tasks are *aligned* (Sun et al., 2020). In prior work, task alignment has been an art, combining ingenuity with trial and error (Gandelsman et al., 2022; Wang et al., 2023). Crucially, the amount of ingenuity in task design does not scale with more data and compute. Our main approach is to learn an aligned self-supervised task from data, instead of handcrafting it from human priors. Specifically, we learn a self-supervised task such that TTT on it actually improves performance on the main task.

Since TTT already defines a learning problem, learning its self-supervised task is a form of *learning to learn*, *i.e.* meta-learning or bi-level optimization (Schmidhuber, 1987). The literature refers to the two nested learning problems as the inner and outer loop. At training time, the *inner loop* learns with self-supervision on each training instance individually, as if it were a test instance. The *outer loop* learns to align the self-supervised task with the main task on the entire training set. At test time, we only invoke the inner loop, *i.e.* TTT. We name our algorithm MTTT, with M for meta.

To better understand MTTT, we look at its simplest nontrivial instantiation, where all components are linear models, and the inner loop takes only one gradient step. Given fixed outer-loop parameters, the inner loop turns out to be equivalent to forward inference with linear attention, *i.e.* self-attention without softmax (Katharopoulos et al., 2020). For a linear transformer, *i.e.* transformer with only linear attention layers, we can replace each with an inner loop. Nesting multiple such inner loops into one outer loop, the most naive case of MTTT is equivalent to training a linear transformer.

It also turns out that our inner loop with a particular kernel estimator is theoretically equivalent to self-attention (with softmax), so MTTT with multiple such inner loops is equivalent to training a transformer. This suggests that our framework is compatible with existing, successful architectures.

To extend beyond existing equivalences, we investigate TTT with neural networks. This performs much better than TTT with linear models (*i.e.* linear transformers), in settings where transformers run out of memory and time. Given the freedom inside our inner loop, we can augment it with heuristics like output normalization and stochastic gradient descent that improve results even more.

Our inner loop *mirrors* regular (non-meta) learning in design, because it breaks each instance into pieces, *i.e.* tokens, that are explicitly treated as data. This perspective is further validated by our empirical evidence, which is not explained through any existing perspective for architecture design. Given the historic success of deep learning over kernels and linear models, we conjecture that such success can potentially be replicated in our inner loop, with more compute and data under MTTT.

## 2 INNER LOOP: TEST-TIME TRAINING WITH RECONSTRUCTION

The architecture for TTT has a shared feature extractor with two output heads. The self-supervised task has a head $g$, and the main task has a head $h$. At test time, the model can only learn from the self-supervised task, so the heads share a feature extractor $f$. This way, TTT can update the shared features, thus helping the main task if it uses the same kind of features as the self-supervised task. Altogether, this architecture looks like the letter 'Y', where $f$ is the stem, $g$ and $h$ are the branches.

In principle, TTT is compatible with any choice of self-supervised task. Here we focus on one general-purpose and domain-agnostic family of self-supervised tasks – reconstruction, since it has been highly effective in prior work (Vincent et al., 2008; Pathak et al., 2016; Brown et al., 2020; Bao et al., 2021; He et al., 2021). For reconstruction, the feature extractor $f$ is also known as the encoder, and the self-supervised head $g$ as the decoder; $g \circ f$ together is called an autoencoder.

Following a standard process called tokenization, each instance is always broken into a sequence of $n$ tokens, so we denote both the instance and sequence by $X = (x_1, \ldots, x_n)$, with token $x_i \in \mathbb{R}^d$.[1] Our basic unit of reconstruction is each individual token $x_i$. The reconstruction target is $x_i$ itself, but the input is transformed by a given function $\phi$, such as adding noise (Vincent et al., 2008) and random masking (He et al., 2021). For each $X$, we optimize the parameters of $f$, denoted by $W$. Overall, the self-supervised loss is

$$\ell(W; X) = \frac{1}{2n} \sum_{i=1}^{n} \left\| g \circ f\left(\phi(x_i); W\right) - x_i \right\|^2. \tag{1}$$

Note that the decoder $g$ is also considered given within the scope of TTT, which only updates $W$.[2] Optimization is performed with $T$ gradient steps. For each $t = 1, \ldots, T$,

$$W_t = W_{t-1} - \eta \nabla \ell(W_{t-1}; X), \tag{2}$$

where the initial value $W_0$ and the learning rate $\eta$ are given, like $\phi$ and $g$.

For the main task, we also transform its input $x_i$ by a given function $\psi$, in the spirit of symmetry to $\phi$ for the self-supervised task. In prior work, $\psi$ has mostly been the identity transform, but Section 3 will make $\psi$ nontrivial, adding expressiveness to the outer loop. Next, we produce the main task outputs by applying $h \circ f$ individually on each $\psi(x_i)$. For convenience, we overload $h, f$ and $\phi$ so they can produce an output sequence from an input sequence:

$$X_{\text{out}} = h \circ f\left(\psi(X); W_T\right) = \left( h \circ f\left(\psi(x_1); W_T\right), \ldots, h \circ f\left(\psi(x_n); W_T\right) \right). \tag{3}$$

Equation 3 could be the last step for main tasks that require $n$ predictions (*e.g.* language modeling), but for other tasks that require a single prediction (*e.g.* object recognition), it is standard to apply an aggregation function across the output sequence, predicting $\hat{y} = \texttt{aggregate}(X_{\text{out}})$ in the end.

---

[1]To be precise, $x_i \in \mathbb{R}^d$ is actually the token's embedding, not the token itself. For $X$ a paragraph of text, each token is usually a (sub-)word; for $X$ an image, each token is usually a patch or pixel. While the type of tokens can potentially be non-numeric, standard techniques are available to embed them into vectors.

[2]While the decoder $g$ also contains learnable parameters, we do not optimize them during TTT in this paper. Our choice, although nonstandard for autoencoders, makes learning to learn conceptually easier in Section 3. Moreover, Sun et al. (2020) and Gandelsman et al. (2022) have shown that whether or not $g$ is optimized during TTT makes little empirical difference. In fact, for $T = 1$ (using notations defined for Equation 2), whether or not a gradient step is taken on $g$ does not matter at all, because $g$ affects the final prediction only through $W_1$.

## 2.1 CONTEXT WINDOW AS A DATASET

In standard terminology, $X = (x_1, \ldots, x_n)$ is called the context window, and $n$ the window length. But for TTT, $X$ is a dataset of size $n$, where each token $x_i$ is actually a non-independent and non-identically distributed piece of data. This intuition is consistent with our algorithm: Equation 1 simply sums the losses individually across tokens, just like across pieces of data; Equation 3 also processes each $x_i$ individually as a "test token", like how a fixed model processes each test instance.

Tokenization enables us to reuse $f$ on $n$ different parts (tokens) of $X$, by treating them as pieces of data, and $X$ as a dataset. It brings the units of operation for TTT "one level below" their traditional sense in machine learning, where $X$ is a piece of data, and a collection of $X$s is a dataset. TTT can be applied without tokenization, but then $X$ would be singleton, unless augmentations are used to create an artificial batch like in Sun et al. (2020).

## 3 OUTER LOOP: LEARNING THE SELF-SUPERVISED TASK FOR TTT

As noted above, TTT does not modify the initialization $W_0$ for encoder $f$, the transformations $\phi$ and $\psi$, or the decoder $g$ and main task head $h$. Altogether, these important components must be determined outside of the scope of TTT. Prior work has tried various heuristics, discussed in Subsection 6.2. Here we take the more principled approach of directly optimizing the final prediction loss on the main task after $T$ steps of TTT.

We first explicitly express the learnable parameters that were hidden in Section 2 because they were considered given within the scope of the inner loop. These are the parameters of $g, h, \phi$ and $\psi$, denoted by $\theta_g, \theta_h, \theta_\phi$ and $\theta_\psi$. We group them together with $W_0$ into $\boldsymbol{\theta} = (\theta_g, \theta_h, \theta_\phi, \theta_\psi, W_0)$, since they will all be learned in the outer loop. Technically, $\boldsymbol{\theta}$ should also contain the learnable parameters of `aggregate`, which we omit for convenience.

Now we derive the outer-loop objective $\mathcal{L}_T$. Denote the main task loss by $\mathcal{L}$, *e.g.* the cross-entropy loss. In the trivial case, for $T = 0$, *i.e.* without TTT, the final prediction loss is exactly $\mathcal{L}$. To be precise, for each instance $X$ with unknown label $y$,

$$\mathcal{L}_0(\boldsymbol{\theta}; X, y) = \mathcal{L}(h \circ f(\psi(X); W_0), y). \tag{4}$$

For $T = 1$, the parameters of $f$ become $W_1 = W_0 - \eta\nabla\ell(W_0; X)$, as defined in Equation 1. Therefore, the final prediction loss for the main task is

$$\mathcal{L}_1(\boldsymbol{\theta}; X, y) = \mathcal{L}(h \circ f(\psi(X); W_1), y) = \mathcal{L}(h \circ f(\psi(X); W_0 - \eta\nabla\ell(W_0; X)), y). \tag{5}$$

For any $T \geq 1$, $\theta_g$ and $\theta_\phi$ implicitly determine the inner-loop loss function $\ell$ defined in Equation 1, therefore affect $\mathcal{L}_T$ through $\nabla\ell$. In other words, $\theta_g$ and $\theta_\phi$ parameterize the self-supervised task.[3] Going further, for $T \geq 2$,

$$\mathcal{L}_T(\boldsymbol{\theta}; X, y) = \mathcal{L}(h \circ f(\psi(X); W_T), y) \tag{6}$$

would be cumbersome to write out in terms of $W_0$, but can be expressed recursively, with $W_t$ defined in Equation 2 for each $t = 1, \ldots, T$.

At training time, the outer loop calculates $\mathcal{L}_T$ individually for each labeled training instance $X$, then optimizes the average $\mathcal{L}_T$ on the entire training set with (a variant of) stochastic gradient descent. Calculating $\nabla\mathcal{L}(\boldsymbol{\theta}; X, y)$ requires taking gradients through $\nabla\ell(W_t; X)$ for $t = 0, \ldots, T-1$., since the latter is implicitly a function of $W_0, \theta_g$ and $\theta_\phi$. This turns out to be easily programmable in JAX, and surprisingly efficient in practice, as we will show in Section 5.

## 4 CHOICE OF LEARNER FOR INNER LOOP

While our inner loop is a sequence of forward and backward operations, it can also be represented as a single forward operation on its unrolled computation graph, so the outer loop becomes regular (non-meta) learning using this graph as a fixed model. It turns out that for simple choices of the inner-loop learner, this equivalent graph can be interpreted through the lens of architecture design.

---

[3]Note that even though $\theta_g$ and $\theta_\phi$ are included as arguments of $\mathcal{L}_T$ for all values of $T$, they do not actually matter for $\mathcal{L}_0$. When the inner loop is trivial, *i.e* runs for 0 iteration, learning to learn collapses to regular (non-meta) learning, and the self-supervised task does not matter.

### 4.1 TTT with Linear Models: Equivalence to Linear Attention

The simplest choice for the feature extractor $f$ is a linear model:

$$f(x; W) = Wx. \tag{7}$$

And the outer-loop components $g$, $h$, $\phi$ and $\psi$ are linear as well. Specifically,

$$g(x; \theta_g) = \theta_g^T x, \quad h(x; \theta_h) = \theta_h x, \quad \phi(x; \theta_\phi) = \theta_\phi x, \quad \psi(x; \theta_\psi) = \theta_\psi x. \tag{8}$$

To make the math even simpler, we always initialize the feature extractor with $W_0 = 0$. Under this construction, the self-supervised loss in Equation 1 becomes

$$\ell(W; X) = \frac{1}{2n} \sum_{i=1}^{n} \|g \circ f(\phi(x_i); W) - x_i\|^2 = \frac{1}{2n} \sum_{i=1}^{n} \|\theta_g^T W \theta_\phi x_i - x_i\|^2. \tag{9}$$

For $W_0 = 0$, one gradient step with learning rate $\eta = 1$ produces

$$W_1 = W_0 - \nabla \ell(W_0; X) = \frac{1}{n} \sum_{i=1}^{n} (\theta_g x_i)(\theta_\phi x_i)^T. \tag{10}$$

Using $W_1$ as the updated weights for the feature extractor, the updated features for each token $x_j$, $j = 1, \ldots, n$, becomes

$$f(\psi(x_j); W_1) = \frac{1}{n} \sum_{i=1}^{n} (\theta_g x_i)(\theta_\phi x_i)^T \theta_\psi x_j. \tag{11}$$

This happens to be linear attention (explained in Appendix A), where $\theta_\phi$, $\theta_\psi$, $\theta_g$ are the key, query, value weights. $h$ is the projection operation used for multi-head attention, discussed in Appendix B.

### 4.2 TTT with Kernels: Equivalence to Self-Attention

So far, we have considered $f$ with explicit parameters. But machine learning is more than just parametric models and gradient-based optimization. Here we consider $f$ as a non-parametric learner.

Recall that non-parametric learning produces an algorithmic function controlled by the training data $x_1, \ldots, x_n$, without explicit parameters of a fixed shape. So our notation for the encoder changes from $f(x; W)$ to $f(x; x_1, \ldots, x_n)$. For example, the nearest neighbor $f(x; x_1, \ldots, x_n)$ simply looks for the most similar piece of training data. Some other non-parametric learners are: support vector machines (SVMs), radial basis function networks, and kernel ridge regression.

But unlike most cases of non-parametric learning, our data for TTT come without labels, since $x_1, \ldots, x_n$ are just tokens of an unlabeled test instance $X$. Analogous to parametric learners, non-parametric ones can also learn with self-supervision to produce better features for a main task downstream. So for each $i = 1, \ldots, n$, we create each label $z_i = \theta_V x_i$ from the unlabeled input $x_i$ itself, where $\theta_V$ is an outer-loop parameter like $\theta_g$ in the parametric case.

The popular self-attention (with softmax) is equivalent to TTT with $f$ as the time-honored Nadaraya-Watson estimator (Bierens, 1988; Cai, 2001), which outputs a locally weighted average of labels $z_i$, $i = 1, \ldots, n$, using a kernel $\kappa$ as the weighting function:

$$f(x; x_1, \ldots, x_n) = \frac{1}{\sum_{i=1}^{n} \kappa(x, x_i)} \sum_{i=1}^{n} \kappa(x, x_i)\, z_i. \tag{12}$$

See Appendix C for a detailed derivation of this estimator. We choose the kernel $\kappa$ to be

$$\kappa(x, x'; \theta_K, \theta_Q) \propto e^{(\theta_K x)^T \theta_Q x'} \tag{13}$$

where $\theta_K$ and $\theta_Q$ are known as bandwidth hyper-parameters for kernels. But for MTTT, they are outer-loop parameters like $\theta_V$. As detailed in Appendix C, asymmetric kernels like our $\kappa$ above have enjoyed a long tradition (Breiman et al., 1977; Chen, 2017). Altogether, Equation 12 and 13 combined is the same as self-attention, where $\theta_K, \theta_Q, \theta_V$ are the key, query, value weights.

Unlike the parametric case, TTT with kernels does not solve an optimization problem, therefore does not produce a different implementation from self-attention. While our equivalence here only provides an alternative interpretation, the fact that both linear models and kernels are empirically effective as inner-loop learners suggests that other learners might also be effective.

### 4.3 TTT with Neural Networks

From the past three decades of progress in machine learning, we observe that the performance of

$$\textit{deep learning} \; > \; \textit{kernels} \; > \; \textit{linear models}$$

given enough data and compute. In Subsection 2.1, we discussed the perspective that our inner loop mirrors regular (non-meta) learning, at least in terms of algorithmic design. To collect empirical evidence for this perspective, we investigate if the ordering above is preserved within our inner loop.

It is well known that transformers with self-attention (TTT with kernels) often outperform those with linear attention (TTT with linear models), *i.e.* linear transformers (Katharopoulos et al., 2020). This validates the rightmost link of the ordering within our inner loop. But TTT with neural networks has no existing equivalence, so we devote the rest of the paper to taking a small step in this huge search space. We delay implementation details such as architecture and optimization to Section 5, and end this subsection with one remaining conceptual implication.

TTT with neural networks and linear models, or any parametric learner, has complexity linear in $n$ for each test instance $X = (x_1, \ldots, x_n)$, since complexity for each token is constant in $n$, and only proportional to the number of parameters. TTT with any non-parametric learner, however, cannot have linear complexity by definition, since its complexity for each token cannot be constant in $n$, *i.e.* amount of training data. For Nadaraya-Watson, complexity for each token happens to be linear. This serves as an alternative explanation for the quadratic complexity of self-attention.

## 5 Experiments

The goal of our experiments is not to be the top on leaderboards, but to evaluate our key perspective, that the inner loop mirrors regular (non-meta) learning, in terms of three qualities. 1) *Descriptive*: Does our equivalence to linear attention hold in practice? 2) *Prescriptive*: Does our perspective show a path for new methods with better performance? 3) *Predictive*: Does our perspective accurately explain the empirical behaviors of new methods?

**TTT layers.** The cleanest and most practical way to answer these questions is to replace every attention layer in an architecture with a TTT inner loop, because ultimately, attention layers are only used as parts of an architecture. Since the inner loop here functions as a *drop-in replacement* for attention, we call it a *TTT layer*, which can also be thought of as an equivalent computation graph (discussed in Section 4). After dropping in the TTT layers, the entire architecture can be trained with MTTT, using the same recipe as that with attention layers, without TTT.

**Variants of MTTT.** We call our method *MTTT-Linear* when encoder $f$ is linear in each TTT layer, and *MTTT-MLP* when $f$ is a multi-layer perception (MLP). We always keep $g, h, \phi, \psi$ linear following Subsection 4.1. For MTTT-Linear, we always keep $W_0 = 0$ fixed to ensure equivalence to linear attention, since MTTT-Linear is only used to investigate descriptiveness. For MTTT-MLP, we experiment with the two design choices below, to investigate the prescriptive power of our perspective. For simplicity, we always set the inner-loop learning rate $\eta = 1$.

**Inner-loop architecture.** For MTTT-MLP, the MLP architecture simply follows standard design in transformers. Concretely, our MLP has 2 linear layers with GELU activation in between; the input and output dimension are the same, and the hidden dimension is $4\times$ as large. The only architectural change, called *Decoder LN*, is that we add a layer norm (LN) after the output of $g$, to normalize the reconstruction outputs, in the spirit of He et al. (2021). We explain this design choice in Figure 2, deferred to the appendix due to space constraints.

**Inner-loop optimization.** When the inner loop takes $T > 1$ steps, each gradient step, by default, uses the average loss over all the tokens, defined in Equation 1. But $T$ steps make the inner loop $T\times$ slower. Given the popularity of stochastic gradient descent (SGD) in deep learning, we use it for our inner loop. Specifically, we randomly split the $n$ tokens into $T$ mini-batches, each of size $T/n$, and take one inner-loop step per mini-batch. Therefore, $T$ steps of SGD combined consumes the same amount of compute as a full-batch gradient step over all the $n$ tokens together.

| Drop-in layer | Acc. (%) | Params. (M) | FLOPs |
|---|---|---|---|
| Linformer (Wang et al., 2020b) | 71.9 | 22.2 | 0.9× |
| Longformer (Beltagy et al., 2020) | 76.3 | 27.4 | 1.1× |
| SOFT (Lu et al., 2021) | 74.6 | 23.5 | 0.9× |
| Hyena (Poli et al., 2023) | 74.8 | 23.5 | 1.0× |
| Self-attn. (Beyer et al., 2022) | 76.5 | 22.1 | 1.1× |
| Linear attn. (Katharopoulos et al.) | 73.2 | 22.1 | 1.0× |
| Linear attn. identity map | 73.0 | 22.1 | 1.0× |
| MTTT-Linear | 72.8 | 22.1 | 1.1× |
| MTTT-MLP | 74.6 | 24.6 | 1.5× |

Table 1: Results on ImageNet. FLOPs are presented as relative to linear attention. Our inner-loop dataset is tiny, with $n = 196$. MTTT-Linear matches linear attention with identity map, as expected. MTTT-MLP outperforms both by a nontrivial margin, but is $1.5\times$ slower than linear attention. Also as expected, self-attention, *i.e.* the original ViT performs the best. See Subsection 5.2 for details.

## 5.1 IMAGENET

We first experiment with the standard setting of ImageNet object recognition (Deng et al., 2009). Our benchmark architecture is Vision Transformer (ViT) (Dosovitskiy et al., 2020). We adopt the well-known recipe of Beyer et al. (2022) by the ViT authors, and their recommended setup for fast research turnaround – training ViT-Small for 90 epochs. With an accuracy of 76.5%, it is often regarded as a fast and competitive baseline. Its recipe splits each image into $14 \times 14$ patches, then embeds each patch with a learned projection. So each $X$ becomes $n = 196$ tokens.

Thinking of the context window as training data for TTT, a dataset of size 196 is not nearly enough for deep learning, if adequate for a linear model. Since over-parameterized neural networks are known to be able to regularize themselves (Zhang et al., 2021), MTTT-MLP should not do poorly, but might not justify the extra compute. In addition, small $n$ means our linear complexity is less of an advantage, in comparison to self-attention (with softmax).

Our results in Table 1 confirm those expectations. MTTT-MLP outperforms MTTT-Linear by a small margin, but uses more FLOPs. If MTTT-MLP was using a smaller architecture that matches the FLOPs of MTTT-Linear, it would have performed worse. Self-attention, for which the training recipe was originally designed, performs the best.

In terms of descriptiveness, MTTT-Linear almost exactly matches linear attention (identity map) – the 0.2% difference is likely due to random noise and loss of numeric precision. However, MTTT-Linear uses $0.1\times$ more FLOPs as linear attention. This extra factor exists because the JAX compiler is unaware that the compiled inner loop will receive $W_0 = 0$ so all those terms involved can be eliminated. We manually calculated the total number of FLOPs for those terms involving $W_0$, and found that it matches the difference in FLOPs between MTTT-Linear and linear attention.

Taking more gradient steps in the inner loop significantly improves accuracy of MTTT-MLP up to $T = 4$, as shown in the left panel of Figure 1. However, $T$ steps on the full batch costs $T\times$ number of FLOPs. So this improvement is predictive but not practically useful. We have experimented with SGD and found that it does not help here. Since $n = 196$ is already a small batch size, splitting 196 tokens into even smaller mini-batches for SGD is usually considered bad practice for deep learning.

The right panel of Figure 1 shows the average $\ell(W_t; X)$ across the test set, for TTT layer 6 (out of 12 in total). The plot for all layers is deferred to Figure 3 in the appendix due to space constraints, but the overall behavior is essentially the same across layers. The five lines are for $t = 0, \ldots, T$, where $T = 4$, *i.e.* the optimal choice of $T$ according to the left panel. For every epoch of outer-loop learning, average inner-loop loss decreases monotonically with more steps. The behavior of this novel inner loop matches that of regular learning with successful optimization.

While MTTT has not been practically useful in this setting, its behavior matches our expectations, indicating that our perspective is predictive on top of descriptive. Note that every hyper-parameter

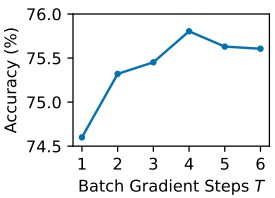 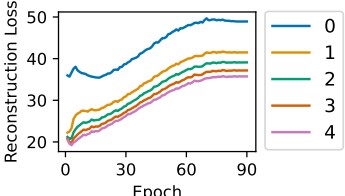

| Dec. LN | Train $W_0$ | Acc. (%) |
|:---:|:---:|:---:|
| ✗ | ✗ | 72.9 |
| ✗ | ✓ | 73.0 |
| ✓ | ✗ | 73.8 |
| ✓ | ✓ | 74.6 |

Figure 1: More inner-loop steps improve accuracy up to $T = 4$ (left). Behavior of inner-loop loss mirrors regular (non-meta) learning (right).

Table 2: Ablations on ImageNet. See Subsection 5.1 for details.

is set according to Beyer et al. (2022), and we have not changed any to get the expected behavior. Our inner-loop learning rate $\eta$ has always been 1, derived from equivalence to linear attention.

In Table 2, we ablate MTTT-MLP with the four combinations of whether or not to use Decoder LN and train $W_0$ in the outer loop. We choose these two factors since Decoder LN is our own design, and training $W_0$ goes a step further from equivalence to linear attention, which requires fixing $W_0 = 0$. Empirically, both components prove to be important for good performance. Therefore, we always keep them for future experiments, without spending more resources to ablate them.

For additional context around our results, we run a few baselines that also have linear complexity. Linear attention as proposed by Katharopoulos et al. (2020) uses manually engineered features of the input tokens, instead of the input tokens themselves. We label the former with citation, and the latter with *identity map*. Other baselines have roughly the same accuracy as MTTT-MLP. Longformer stands out with the same accuracy as self-attention, but we find that the default window size for its sliding attention is $512 > 196$, so it happens to be the same as self-attention for $n = 196$.

## 5.2 IMAGENET FROM $224 \times 224$ RAW PIXELS

To better evaluate our perspective that the inner loop mirrors regular (non-meta) learning, we need a setting where the sequence length $n$, *i.e.* amount of training data for the inner loop, is actually comparable to the amount in typical applications of deep learning. Inspired by Chen et al. (2020), we experiment with ImageNet object recognition using raw pixels instead of patches as input tokens. This gives us $n = 224 \times 224 = 50,176$.

For Chen et al. (2020), the point of using pixels is to eliminate image-specific prior knowledge.[4] At a high level, the progress in deep learning over the past decade can be seen as gradually eliminating human priors, in favor of general methods that take advantage of data and compute. Following their setting, we use learned positional embeddings, instead of engineered positional encoding. Therefore, our entire system is permutation invariant.

While Chen et al. (2011) do not use any data augmentation, they use a much larger collection of images. We have been able to remove the augmentations except one – random resize crop (Szegedy et al., 2015), without which all methods fail to get more than 40% accuracy. Since random resize crop does not add any synthetic artifact to natural images, we justify it as using more data without actually using another dataset. We always use random resize crop for the rest of the subsection.

Experiments in this subsection are conducted with ViT-Tiny unless noted otherwise, because training with 50k tokens per instance is very compute-intensive. Every other aspect of our recipe follows Beyer et al. (2022), like in Subsection 5.1. Our results are in Table 3. Self-attention, which performed the best with patches, cannot fit in memory. Even if memory was not an issue, it would still need at least $200\times$ more FLOPs than linear attention according to our estimations.

We highlight two results. First, taking $T = 4$ steps of SGD improves accuracy by 3.3% on top of MTTT-MLP with $T = 1$, without costing extra FLOPs. To the best of our knowledge, this improvement cannot be explained through any existing perspective without an explicit inner loop.

---

[4]While transformers have already eliminated the locality prior in convolutions, most papers on ImageNet still use patches instead of pixels as input tokens. This is equivalent to a first layer of convolutions where the filter size and stride size both equal to the patch size, and is in fact often implemented as such. Using raw pixels as input tokens eliminates locality prior completely.

| Model | Drop-in layer | Acc. (%) | Params. (M) | FLOPs |
|---|---|---|---|---|
| ViT-Tiny | Self-attn. (Beyer et al., 2022) | - | 5.6 | 200× |
| | Linear attn. (Katharopoulos et al.) | 53.7 | 5.6 | 1.0× |
| | Linear attn. identity map | 49.9 | 5.6 | 1.0× |
| | MTTT-Linear | 50.0 | 5.6 | 1.1× |
| | MTTT-MLP | 61.9 | 6.8 | 1.8× |
| | MTTT-MLP SGD $T = 4$ | 65.2 | 6.8 | 1.8× |
| ViT-Small | Linear attn. (Katharopoulos et al.) | 54.4 | 21.8 | 3.9× |
| | Linear attn. identity map | 55.7 | 21.8 | 3.9× |

Table 3: Results on ImageNet from pixels. FLOPs are presented as relative to linear attention. MTTT-MLP with SGD outperforms without by 3.3%, and does not cost extra FLOPs. It improves almost 10% on top of a ViT-Small with linear attention, which uses more than $3\times$ parameters and $2\times$ FLOPs. See Subsection 5.2 for details.

Like in Figure 1, our inner-loop loss with SGD steps also behaves like regular learning, as shown in Figure 4 of the appendix. Second, MTTT-MLP with SGD improves almost 10% on top of even a ViT-Small with linear attention, which uses more than $3\times$ parameters and $2\times$ FLOPs. For SGD, $T = 4$ was simply chosen according to the optimal on patches.

These pieces of empirical evidence indicate that our perspective is prescriptive, by showing a path to new methods with better performance. It is also predictive, since expectations derived from regular learning accurately explain novel behaviors of the inner loop, without any hyper-parameter tuning. In terms of descriptiveness, MTTT-Linear matches linear attention (identity map) within 0.1%.

## 6 RELATED WORK

### 6.1 IN-CONTEXT LEARNING AS EXPLICIT LEARNING

To the best of our knowledge, three pieces of prior work (Akyürek et al., 2022; Dai et al., 2022; Von Oswald et al., 2023) have independently proposed the idea that linear transformers can simulate some variant of linear regression on in-context data, as an explanation for in-context learning. Take Von Oswald et al. (2023) as an example. Given a labeled dataset, their work first trains a linear regression model with $T$ gradient steps, then constructs the weights of a $T$-layer linear transformer to produce the same output as the trained linear model.

Our work differs in two main aspects: self-supervision and direction of claims. First, prior work focuses on showing that (linear) transformers can simulate learning on specific, supervised objectives, *e.g.* ridge regression, so their constructions rely on labeled pairs of in-context training data. If there is a meta-learning component, it is restricted to specific hyper-parameters, *e.g.* the learning rate. On the other hand, our inner loop implements a general objective that itself is mostly learned, so it does not need labeled data. This makes our inner loop less interpretable but more practical.

At a higher level, transformers are complex models, and linear models are simple. Prior work uses the complex to construct the simple. Our construction takes the converse direction. In prior work, empirical performance of meta-learning with linear regression has been significantly worse than linear transformers, even on labeled in-context data. Again, with the goal of explaining transformers, their claims often indicate that linear transformers are superior to meta-learning. Our experiments also point towards the converse.

Recently, Mahankali et al. (2023); Zhang et al. (2023); Ahn et al. (2023) and Tarzanagh et al. (2023) have further extended the arguments in prior work, therefore inheriting their two aspects above. Tarzanagh et al. (2023), in particular, argues that transformers implement non-parametric learners (SVMs) on labeled data, supporting our intuition in the converse direction. In summary, our paper complements prior work, with the different goal of inspiring potentially more powerful systems.

## 6.2 LEARNING AT TEST TIME

The idea of learning at test time has a long history in machine learning. One of the earliest instantiations of this idea is Bottou & Vapnik (1992): For each test input, train on its neighbors before making a prediction. This idea continues to be effective for SVMs (Zhang et al., 2006) and large language models (Hardt & Sun, 2023). In computer vision, the general idea of learning at test time has also been applied to specific applications (Jain & Learned-Miller, 2011; Shocher et al., 2018; Mullapudi et al., 2018; Luo et al., 2020; Nitzan et al., 2022).

*Transductive learning* (Gammerman et al., 1998) is the first to articulate our philosophy in Section 1. As stated by Vapnik (2013): "Try to get the answer that you really need, but not a more general one." Implementation-wise, it uses test data to add constraints to the margin of SVMs (Joachims, 2002; Collobert et al., 2006). This is an example of non-parametric learning at test time, similar to our kernel estimator in Subsection 4.2. However, transductive learning usually needs multiple test instances to be practically effective, unlike our method, which only needs a single instance at a time.

Next we have an in-depth discussion of two particular relevant lines of work: TTT and fast weights.

### 6.2.1 TEST-TIME TRAINING WITH SELF-SUPERVISION

Our inner loop performs TTT with self-supervision, discussed in Section 2. This general framework was first proposed by Sun et al. (2020), with results for supervised learning under distribution shifts. Unlike previous lines of work, TTT can be used in principle with any self-supervised task, on any type of data, for any application, making it particularly suitable for deep learning. Follow-up work has applied TTT to batches of data (Wang et al., 2020a; Liu et al., 2021), and other main tasks like robot manipulation (Hansen et al., 2020) and locomotion (Sun et al., 2021), among others.

Particularly relevant to our inner loop, Gandelsman et al. (2022) performs TTT with reconstruction as the self-supervised task, and Wang et al. (2023) applies this method online to video streams. The biggest difference is that our reconstruction task is parameterized for meta-learning. In addition, our inner loop obtains multiple units of learning, $x_1, \ldots, x_n$, out of a single test instance through tokenization. In prior work, each unit of learning is created through either data augmentations or a randomized $\phi$, such as masking random patches (He et al., 2021).

### 6.2.2 FAST WEIGHTS

The general idea of *fast weights* is to update the parameters of a "fast" model on the most relevant data, as opposed to a "slow" model on all data (Hinton & Plaut, 1987; Tieleman & Hinton, 2009), which most people today simply refer to as training or learning. The most relevant data can be the test instance itself, where the update is performed without human supervision at test time. Our work shares the same general idea, but formulates an explicit learning problem for each inner-loop update, with the goal of generalizing to that test instance.

To make fast weights "fast", *i.e.* efficient, their update rules avoid forming an optimization problem with explicit objectives on the training data, *i.e.* a learning problem. For example, given each input $x$, one popular update rule for fast weights is to add $xx^T$ (or some variant thereof) (Ba et al., 2016) like in Hebbian learning and Hopfield networks (Hopfield, 1982). In contrast, our update rule for TTT is an explicit training process as its name suggests.

*Fast weight programmers* (FWPs) (Schmidhuber, 1992) produce the updates to fast weights with a "slow" model. MTTT's outer loop can be seen as training the "slow" model, if its inner loop is viewed as updating fast weights. In particular, FWPs with the Hebbian update rule above are equivalent to linear transformers (Schlag et al., 2021), therefore also to MTTT with linear models. Clark et al. (2022) add a final layer of fast weights to a transformer and train its initialization with a FWP to improve performance on language modeling.

Given the broadest definition of FWPs, MTTT with parametric models can be seen as a special case (Kirsch & Schmidhuber, 2021). But the difference in update rules between TTT and fast weights, as discussed, carries over to MTTT and FWPs. Irie et al. (2021) have tried "fast" networks with weights directly produced as output of a "slow" network, without forming a learning problem. In contrast, our inner loop mirrors regular (non-meta) learning. This helps us with empirical intuitions like in Figure 1, and heuristics like output normalization and stochastic gradient descent.

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

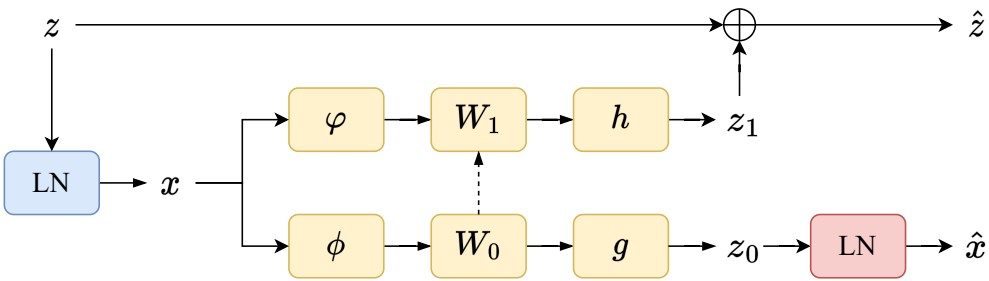

Figure 2: Illustration of Decoder Layer Norm (LN), presented in Section 5. This diagram shows the first half of a transformer block, omitting the second half which does not contain any attention layer. The input embedding is $z$, the output is $\hat{z}$. The identity mapping is at the top, and the residual is learned at the bottom. The dotted line from $W_0$ to $W_1$ represents an inner-loop gradient step. Here we use $T = 1$, *i.e.* only one step in the inner loop, so the final prediction is made with $W_1$. **Standard design**: only use the blue LN. The output of decoder $g$, in this case $z_0$, is expected to reconstruct $x$, causing a "type mismatch". **Our design**: also use the red LN. Now $\hat{x}$ is expected to reconstruct $x$, and both are outputs of LN.

## A    LINEAR ATTENTION AND LINEAR TRANSFORMERS

This section is intended as a very brief reference on linear attention. For a more in-depth discussion, please see Section 3 of Katharopoulos et al. (2020). In this section, we use standard notations for transformers, where $X$ is the $n \times d$ matrix with $x_i$ as its $i$th column. Recall that for self-attention, we first form the keys, queries and values by multiplying $X$ with the respective weight matrices:

$$K = \theta_K X, \ \ Q = \theta_Q X, \ \ V = \theta_V X. \tag{14}$$

Then we obtain the $i$th output embedding $Z_i$ as

$$Z_i = \text{softmax}_{j=1}^n \left( Q_i^T K_j \right) V_j, \tag{15}$$

where softmax makes $Q_i^T K_j$ sum to 1 over $j$. Linear attention simply replaces softmax with mean:

$$V_i' = \frac{1}{n} \sum_{j=1}^n \left( Q_i^T K_j \right) V_j = \frac{1}{n} \sum_{j=1}^n Q_i \left( K_j^T V_j \right) = \frac{1}{n} \sum_{j=1}^n \left( K_j^T V_j \right) Q_i. \tag{16}$$

Since $\sum_{j=1}^n \left( K_j^T V_j \right)$ is the same for each $i$, it can be pre-computed with linear complexity.

Linear attention in Katharopoulos et al. (2020) is slightly different. Before forming the keys, queries and values in Equation 14, it first passes $X$ through an engineered feature transformation (elu + 1). Then for Equation 16, instead of a simple mean, there is a data-dependent normalizer. In Section 5, we label their modified linear attention with citation, and the one described in Equation 14 and 16 as *linear attention with identity map*.

In standard terms, a transformer uses self-attention unless noted otherwise. A linear transformer is simply a transformer with every self-attention layer replaced by a linear attention layer. Our paper follows this convention.

## B    MULTI-HEAD ATTENTION

For an attention layer with $H$ heads, we need $H$ inner loops in parallel. In the case of linear attention, there would be $H$ linear models, each with a weight matrix of size $(d/H) \times (d/H)$ instead of $d \times d$. Under the MTTT perspective, this design naturally forms a bottleneck for compressing information, often critical for autoencoders. Specifically, each $\phi$ now maps from dimension $d$ to $d/H$, and $g$ from $d/H$ back to $d$. Each $h$ here is a projection operation, the *de facto* standard for multi-head attention.

## C    OUR KERNEL ESTIMATOR

Here is the derivation for the Nadaraya-Watson estimator. Throughout this section of the appendix, we use $\mathbf{x}$ to denote the input token $x$ as a random variable, which is different from the test instance (*i.e.* sequence) $X$ in the main text of the paper. Our goal is to produce the corresponding feature, another random variable $\mathbf{z}$. This is formulated as estimating the conditional expectation of $\mathbf{z}$:

$$\mathbb{E}[\mathbf{z}|\mathbf{x} = x] = \int p(z|x)\, z\, dz = \int \frac{p(x,z)}{p(x)}\, z\, dz.$$

Since the true probability distributions $p(x)$ and $p(x, z)$ are unknown, we replace them with their kernel density estimations. Specifically, the kernel density estimation for $p(x)$ is:

$$\hat{p}(x) = \frac{1}{n}\sum_{i=1}^{n} \kappa(x, x_i),$$

where each $x_i$ is a piece of training data in general. (Recall that for our paper, $x_i$ is specifically training data for the inner loop, *i.e.* a token, which matches our notation in the main text.)

For estimating $p(x, y)$, we use the product kernel:

$$\hat{p}(x,z) = \frac{1}{n}\sum_{i=1}^{n} \kappa(x, x_i)\, \kappa'(z, z_i).$$

At first sight, it seem absurd to factor the joint probability into two seemingly independent kernels. But in this case, $\kappa'$ can actually be any $\kappa_i'$ dependent on $x_i$, since it will be integrated out. So the two kernels do not actually need to be independent.

Plugging in those estimations, we obtain the Nadaraya-Watson estimator:

$$
\begin{aligned}
\hat{\mathbb{E}}[\mathbf{z}|\mathbf{x} = x] &= \int \frac{\hat{p}(x,z)}{\hat{p}(x)}\, z\, dz \\
&= \frac{1}{\hat{p}(x)}\int \hat{p}(x,z)\, z\, dz \\
&= \frac{1}{\sum_{i=1}^{n}\kappa(x,x_i)}\int \sum_{i=1}^{n} \kappa(x,x_i)\,\kappa'(z,z_i)\, z\, dz \\
&= \frac{1}{\sum_{i=1}^{n}\kappa(x,x_i)}\sum_{i=1}^{n} \kappa(x,x_i)\int \kappa'(z,z_i)\, z\, dz \\
&= \frac{1}{\sum_{i=1}^{n}\kappa(x,x_i)}\sum_{i=1}^{n} \kappa(x,x_i)\, z_i.
\end{aligned}
$$

Recall that in the main text, our kernel is chosen to be

$$\kappa(x, x'; K, Q) \propto e^{x^T K^T Q x'}, \tag{17}$$

where $K$ and $Q$ have been known as bandwidth hyper-parameters (Williams & Rasmussen, 2006).

In modern days, people think of kernels as positive semi-definite, which might not be guaranteed for $\kappa$ unless $K = Q$. However, people working on kernels decades ago, around the time when the Nadaraya-Watson estimator was popular, have been surprisingly lenient with the choice of kernels. When an estimator uses $Q \neq K$, it is known as a balloon estimator (Chen, 2017). Papers like Breiman et al. (1977) have even used $Q$ as function of $x'$, known as sample-adaptive smoothing.

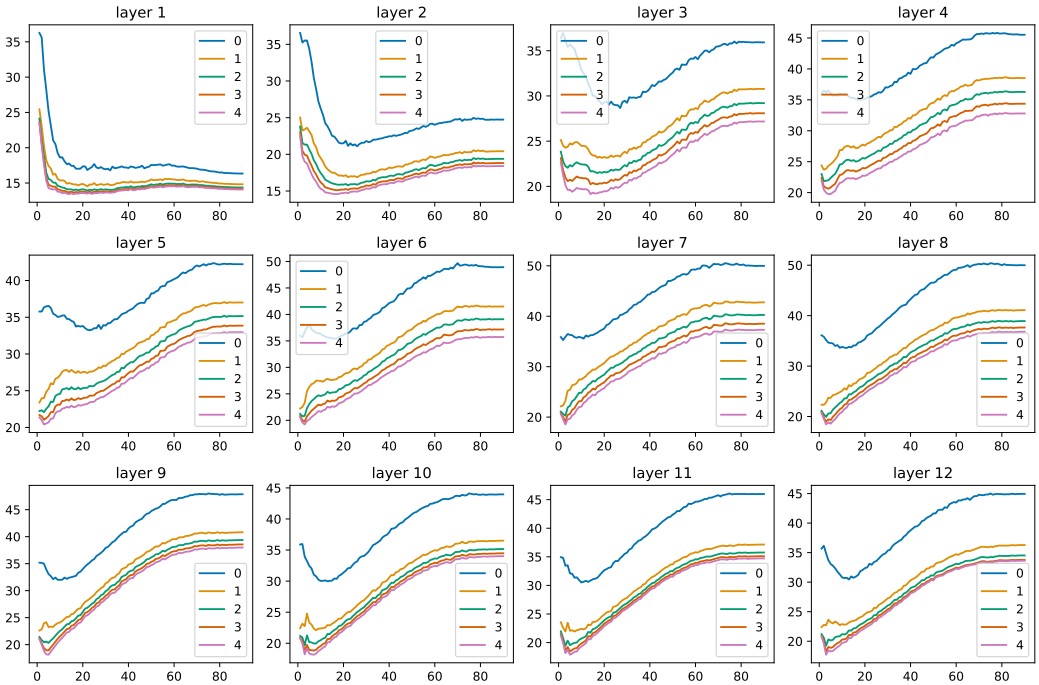

Figure 3: Inner-loop loss across the 12 TTT layers. Behavior across layers is roughly the same as in Figure 1. Method: MTTT-MLP performing full-batch gradient descent in the inner loop, $T = 4$. Setting: ImageNet from patches. See Subsection 5.1.

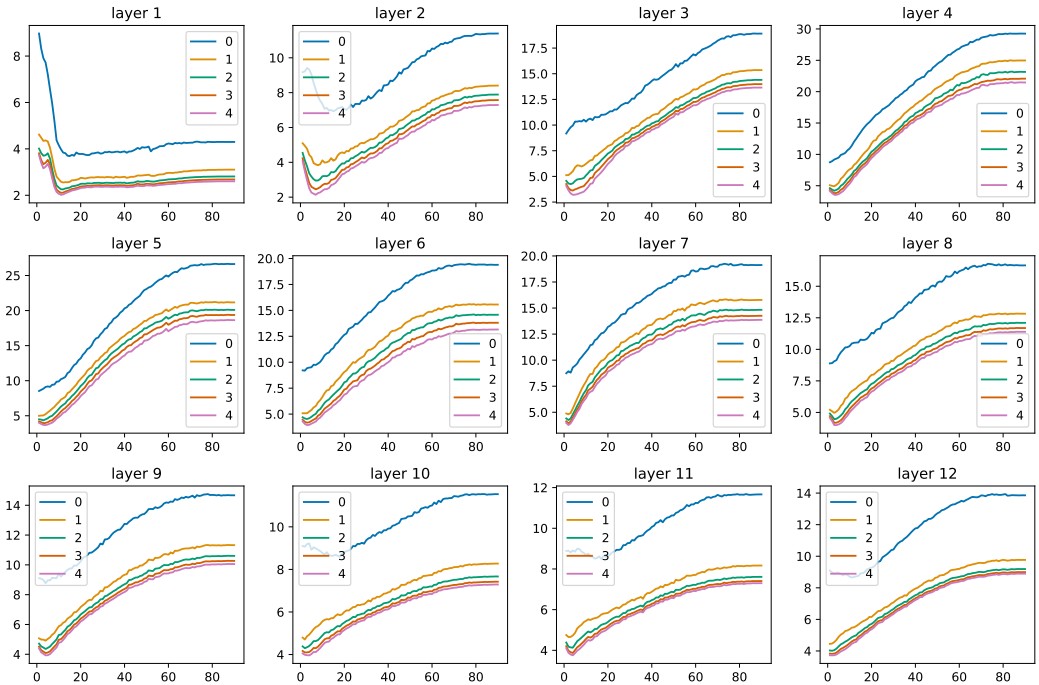

Figure 4: Inner-loop loss across the 12 TTT layers. Behavior across layers is roughly the same as in Figure 1. Method: MTTT-MLP performing stochastic gradient descent in the inner loop, $T = 4$. Setting: ImageNet from pixels. See Subsection 5.2.

