# OpenReview forum: "Learning to (Learn at Test Time)"
_ICLR.cc/2024/Conference — ICLR 2024 Conference Withdrawn Submission_

### Official Review · Reviewer_tf1h · 2023-10-17

**Soundness:** 2 fair
**Presentation:** 3 good
**Contribution:** 2 fair
**Rating:** 3
**Confidence:** 5

**Summary:**

The paper proposes to meta-learn the self-supervised proxy task used in test-time training such that this proxy task is maximally aligned with the actual task. In this perspective, test-time training is also conducted during training and can be considered as the inner-loop of meta-learning, with regular end-to-end learning forming the outer loop. The paper proposes a general framework for this setting (MTTT), and identifies that a minimal all-linear instantiation of MTTT is equivalent to linear attention. The paper studies different variants of the proposed MTTT on ImageNet (patchified and pixel-level).

**Strengths:**

+ The paper nicely motivates the approach of meta-learning the self-supervised proxy task as an alternative to manually tweaking the proxy task (both with the goal of increasing alignment between proxy and actual task)
 + The formal exposition and companying discussion of the meta-learning test-time training framework (MTTT) in Section 2 - 4 is clean, easy to follow, and general
 + The equivalence of linear MTTT to linear attention is nicely identified and sheds some lights into the connection between linear attention and  test-time training
 + The perspective of considering the tokens of a datum to form a non-iid dataset is helpful and refreshing.

**Weaknesses:**

- the novelty of the proposed framework is limited as there exists at least one (uncited) prior work that has substantial overlap: "MT3: Meta Test-Time Training for Self-Supervised Test-Time Adaption", Alexander Bartler, Andre Bühler, Felix Wiewel, Mario Döbler, Bin Yang, Proceedings of The 25th International Conference on Artificial Intelligence and Statistics, PMLR 151:3080-3090, 2022.
- the empirical analysis is limited to image classification on ImageNet (either on patches or on pixels). It would be helpful to consider domain shift settings (e.g. meta-learn on ImageNet, test on ImageNet-C/R etc.) or other tasks (e.g. semantic segmentation, object detection).
- The reviewer is sceptical about using mini-batch SGD in the inner loop because, as discussed by the authors, the tokens are non-iid and it seems suboptimal to perform reconstruction on random subsets of the tokens in isolation.
- while MTTT-Linear is interesting from a theoretical perspective due to its relationship to linear attention, it is otherwise questionable why test-time training would be performed layer-wise and not end-to-end. For instance, reconstruction of masked patches (a common proxy task) is unlikely solvable with a narrow 2-layer MLP (even when unrolled for T iterations). A more natural choice would be to apply test-time training to entire stem at once rather than to individual layers.
- the author states "For example, MTTT-Linear only takes 0.1× more FLOPs than linear attention, but turns out
to be 2× slower in wall-clock time. In principle, the two are equivalent, so such difference can only be explained by systems-level inefficiencies.". It is worth noting the FLOPS and wall-clock time are not equivalent and in many cases FLOPs is a poor proxy for inference latency. Overall, a more thorough empirical investigation of the runtime overhead of the proposed procedures would be helpful for practitioners.
- minor: a graphical illustration of the proposed frameworks on the first pages would make the paper more accessible.

**Questions:**

- can the authors indicate why they did only consider layer-wise MTTT?
- can the authors discuss why the think that mini-batch SGD is applicable for non-iid token datasets in the inner-loop of MTTT?
- can the authors demonstrate the potential of MTTT on other problems than ImageNet classification?
- can the authors clarify the relationship to the prior work MT3 and summarize novelty of MTTT vs. MT3?
Thanks for considering my questions!

---

### Official Review · Reviewer_sCDR · 2023-10-25

**Soundness:** 1 poor
**Presentation:** 1 poor
**Contribution:** 1 poor
**Rating:** 3
**Confidence:** 3

**Summary:**

The objective of this paper, as I understand it, is to meta-learn a reconstruction objective during training, specifically at test time. The authors assert that mastering a self-supervised learning task like reconstruction on individual test instances eliminates the necessity for intricate and deeper transformer architectures, emphasizing the sufficiency of simpler ones.

The authors conduct evaluations on ImageNet using Vision Transformer (ViT) and compare their approach with other self-attention variants, including Linformer and Longformer. This comparison provides valuable insights into the effectiveness of their method within the context of established transformer-based models, contributing to the ongoing exploration of self-supervised learning tasks in the realm of deep learning architectures.

**Strengths:**

S1. The authors tackle the important task of test time training.

**Weaknesses:**

W1: Writing Clarity and Structure:

One of the primary concerns about this paper lies in its writing style. The introduction of the method is abrupt, lacking a detailed discussion of the learning or test-time training setting. The sub-sections appear disjointed, making it challenging to connect concepts such as inner loop and outer loop. Moreover, the excessive use of technical jargon renders the paper inaccessible to outsiders, hindering their understanding. Furthermore, the paper's heavy reliance on NLP motivation does not align with its evaluation on standard image classification—a task less directly related to sequence modeling. The section discussing related work and the positioning of this paper is notably poor, leaving readers, even after multiple readings, struggling to discern the paper's stance and contributions.

W2: Lack of Empirical Benefit and Motivation:

Another significant weakness is the absence of empirical benefits in the proposed approach. The paper fails to demonstrate improvements in accuracy or reductions in the number of parameters or computational FLOPs. While the authors express a lack of interest in leaderboard rankings, the introduction of their MTTT technique lacks a compelling rationale, leaving readers questioning its necessity and relevance. The absence of concrete empirical evidence undermines the paper's persuasiveness and impact, making it crucial for the authors to address this gap in their analysis to strengthen their argument effectively.

**Questions:**

No questions.

---

### Official Review · Reviewer_vvKb · 2023-10-30

**Soundness:** 3 good
**Presentation:** 3 good
**Contribution:** 4 excellent
**Rating:** 6
**Confidence:** 4

**Summary:**

Taking inspiration by test-time training, this paper introduces an instance-adaptive attention mechanism. This mechanism comprises two loops during training. The inner loop involves patch/pixel reconstruction, while the outer loop is dedicated to formal supervised learning. The experiments demonstrate that the proposed method is competitive.

**Strengths:**

1. This is the first time a dynamic neural network has been studied from the perspective of test-time training.

2. The introduced method is very interesting.

3. The experiments show that the proposed method is competitive.

**Weaknesses:**

1. The TTT layer can adapt its parameters to different input data. This is similar to the dynamic neural network [a]. So, it's better to analyze the differences between the proposed one and related dynamic neural networks.
2. To enhance the understanding of this paper, it would be beneficial to include either an algorithm description or a framework figure.
3. Test-time training has been demonstrated to enhance robustness against distribution shifts. The proposed TTT layer adjusts its parameters for each input instance. Therefore, why not compare the results when inputs contain corruptions or distribution shifts, as demonstrated in the experiments conducted in [b]?
4. In the inner loop, the authors employ a reconstruction task as the test-time training method. There are various different schemes for test-time training. Why do the authors choose reconstruction as the training objective specifically?
5. Based on the results in Figure 1 and Table 3, it appears that a smaller reconstruction error leads to better accuracy. However, is this conclusion accurate? I believe it would be beneficial to conduct additional experiments to support that reducing reconstruction error has a positive impact on the final task.
6. I think the TTT layer belongs to a specialized linear attention mechanism, so it's better to compare some recent works like [c].

[a] Dynamic Neural Networks: A Survey. IEEE TPAMI 2021.

[b] Test-time training with self-supervision for generalization under distribution shifts. ICML 2020.

[c] FLatten Transformer: Vision Transformer using Focused Linear Attention. ICCV 2023.

**Questions:**

Apart from the issues mentioned in the Weaknesses section, there are several other concerns:
1. Since the model updates parameters for each instance, what is the inference speed of the proposed method? Does it introduce any additional computational overhead during inference?
2. Comparing Table 1 and Table 3, it is evident that the size of the training set has an impact on the results. The size of the training data is further related to the patch size. So, how does the patch size affect the results?

---

### Official Review · Reviewer_DvfS · 2023-11-04

**Soundness:** 2 fair
**Presentation:** 2 fair
**Contribution:** 2 fair
**Rating:** 3
**Confidence:** 4

**Summary:**

This paper proposed to formulate the test-time training (TTT) problem in a learning-to-learn scheme, i.e., inner-loop and outer-loop optimization.
Specifically, the inner loop involves self-supervised reconstruction tasks for TTT, and the outer loop is a regular supervised learning task.

Under this specific setting, this paper shows that in the simplest case (linear), the update rule for the outer loop ensembles a self-attention mechanism. Then the simplest linear model is replaced with neural networks.

Experiments are performed to verify that the inner loop mirrors regular learning. And various Transformer/self-attention modules are included for comparison.

**Strengths:**

- The overall structure and formulation are reasonable, which puts TTT as the inner loop and supervised training as the outer loop.
- This paper shows that in the simplest linear case, the inner gradient updated to the outer loop, and can be regarded as self-attention.
- Various self-attention modules are compared in the experiment to verify the regular learning hypothesis of the inner loop.

**Weaknesses:**

- (1) While the overall structure of the inner-outer loop is reasonable in this TTT setting, existing works on unsupervised meta-learning/few-shot learning have investigated this inner loop behavior in a similar manner, e.g., [R1-R2]. A fair discussion is lacking.
- (2) This paper motivates from the TTT perspective, but no TTT experiments are performed and compared, e.g., comparing with (Sun et al., 2020).
- (3) All the experiments are about Drop-in layer comparison with various self-attention modules. From the results, the proposed method cannot beat these modules, and the computation cost seems to be larger. I am not sure the practical usage of the proposed method.


[R1] Khodadadeh, Siavash, Ladislau Boloni, and Mubarak Shah. "Unsupervised meta-learning for few-shot image classification." Advances in neural information processing systems 32 (2019).

[R2] Hsu, Kyle, Sergey Levine, and Chelsea Finn. "Unsupervised learning via meta-learning." arXiv preprint arXiv:1810.02334 (2018).

**Questions:**

Please see above.

---

### Author Response · Authors · 2023-11-30

The questions and comments from the three reviewers (DvfS, tf1h, sCDR) giving the rating of 3 indicate that they have not understood the technical content of the paper. The common failure mode is that they cannot disentangle the two orthogonal elements of research: the method and task, colloquially known as the hammer and nail. The method is test-time training (TTT) with learning to learn. The task is simply supervised learning, without so-called distribution shifts.

TTT as a method has often been used for the task of distribution shifts in the last four years. But this correlation does not imply that TTT can only be used for distribution shifts, or that TTT **is the same as** distribution shifts. This confusion between method and task is evident in the reviews:
- “This paper motivates from the TTT perspective, but no TTT experiments are performed” (DvfS)
- “The authors tackle the important task of test time training” (vvKb)

The AC also seems confused, since the primary areas for our paper are: "transfer learning, meta learning, and lifelong learning", even though our paper has nothing to do with "transfer learning" or "lifelong learning". These areas were probably assigned simply as substitues for test-time training, which has not yet become a primary area. Since our paper was assigned to reviewers from these areas, they were probably misled from the beginning to think from that angle. This must have been frustrating for them.

As the author who coined the term test-time training, I would like to apologize for this community-wide confusion. I take responsibility for not clearly stating the distinction between the method (TTT) and task (distribution shifts) four years ago, when I wrote the first paper in this now growing line of work (Sun et al. 2020). TTT has been the only focus of my research since then, and I strongly believe that the community needs to look beyond the task of distribution shifts, for our method - or algorithmic framework - to grow into a more substantial concept.